# Illicit Drug Use and Sociodemographic Correlates Among Adolescents in a Brazilian Metropolitan Region: A School-Based Cross-Sectional Study

**DOI:** 10.3390/ijerph22091373

**Published:** 2025-08-31

**Authors:** Luíza Eduarda Portes Ribeiro, Luisa Sorio Flor, Carlos Augusto Lopes, Franciéle Mabotti Costa Leite

**Affiliations:** 1Program in Public Health, Federal University of Espírito Santo (UFES), Vitória 29043-910, Brazil; franciele.leite@ufes.br; 2Institute for Health Metrics and Evaluation (IHME), University of Washington, Seattle, WA 98121, USA; lsflor@uw.edu; 3Government of the State of Espírito Santo, Sub-Secretariat of Drug Policies, Vitória 29043-910, Brazil; carlopesviana@gmail.com

**Keywords:** illicit drugs, students, adolescent health

## Abstract

(1) Introduction: Drug use among adolescents remains persistent, including in school settings, thus requiring attention. This study analyzed the prevalence of drug experimentation and current use among high school students and their associated factors. (2) Methods: An analytical cross-sectional study was conducted in 2023 with 4610 students from public and private high schools in the Metropolitan Region of Grande Vitória, Brazil. Data were collected using tablets and a questionnaire. Sociodemographic variables were considered. Bivariate analyses and Poisson regression were used. (3) Results: Drug experimentation was reported by 22% of students (23.5% girls; 19.8% boys), and current use by 8.7% (9.2% girls; 8.0% boys). Higher lifetime use was observed among older adolescents (RP: 1.44) and those from higher socioeconomic classes (RP: 1.24). Statistically significant associations were found in LGBTQIAPN+ students (RP: 1.54 experimentation; RP: 1.76 current use) and Black students (RP: 1.33; 1.59). Being in a relationship (RP: 1.41; 1.42), currently working (RP: 1.36; 1.62), and having separated parents (RP: 1.29; 1.37) were also associated. (4) Conclusions: The high prevalence of drug use among adolescents highlights the need for targeted public policies, especially school-based actions promoting mental health, diversity, and racial equity. This study identifies vulnerable subgroups at greater risk.

## 1. Introduction

The use and abuse of psychoactive substances is an increasing global public health concern due to their significant impact on individual and collective health, quality of life, and economic systems. Prevention and treatment of substance misuse are notably included among the United Nations’ Sustainable Development Goals (SDGs) in the 2030 Agenda, reflecting the high costs associated with the rehabilitation of individuals affected by substance use disorders [1,2].

Globally, the 2024 World Drug Report estimated that approximately 292 million people consumed legal or illegal substances in 2022, representing a 20% increase compared to the previous decade [3]. The most consumed drugs were marijuana, opioids, methamphetamines, cocaine, and ecstasy. These substances affect the central nervous system and are classified as depressants, stimulants, or hallucinogens. Depressants, like marijuana and opioids, reduce brain activity, causing euphoria followed by relaxation. Stimulants, such as cocaine, crack, and amphetamines, increase alertness and energy. Hallucinogens, like LSD and ecstasy, cause delusions and hallucinations [4].

Except for some opioids used for therapeutic purposes, these substances are considered illicit in many countries, with their production, distribution, possession, and use being prohibited by law due to their harmful health effects. Moreover, they are often linked to illegal markets and various forms of criminal activity, as defined by national and international legislation [4].

In Brazil, Law No. 1143/2006 established the National System of Public Policies on Drugs (Sisnad), which regulates preventive actions against substance misuse, treatment, and social reintegration of individuals with substance use disorders, as well as repressive measures against trafficking and unauthorized production of psychoactive substances [5]. According to the 2012 National Survey on Alcohol and Drugs (LENAD), between 4% and 6% of Brazilian adolescents had experimented with illicit drugs, with cannabis and cocaine being the most prevalent substances in both adolescents and adults [6].

The seriousness of this issue lies not only in the high potential for chemical dependency but also in the broad and long-lasting repercussions on users’ lives. Drug use is associated with impairments in cognitive and emotional functioning, adverse physical and mental health outcomes, poor academic and professional performance, weakened social and emotional bonds, increased vulnerability to accidents and violence, and a reduced life expectancy, especially among individuals who begin using substances at a young age [3,7,8].

During adolescence and young adulthood, these risks are intensified. This developmental stage, covering individuals aged 10 to 24 years, is marked by significant physical, cognitive, emotional, hormonal, and social changes. The process of identity formation, increased pursuit of autonomy, and peer influence make adolescents particularly susceptible to engaging in risky behaviors such as substance use. Research indicates that the developing brain is more vulnerable to the neurotoxic effects of drugs, increasing the likelihood of early dependence and long-term harm [3,9].

Despite broad recognition of the negative consequences of drug use, data on prevalence and associated factors among adolescents remain limited in many countries. International studies reveal substantial variations: in the United States, the prevalence of illicit drug use among students was 10.2% between 2005 and 2017 [10], whereas in Sweden, it reached approximately 16% [11]. In Brazil, the 2019 National School Health Survey (PeNSE) reported a national average of 13%, with the state of Espírito Santo presenting one of the highest rates at 16% [12]. Considering Brazil’s continental scale, other regions such as the Center-West [13], Northeast [14], and Southeast [15] showed similar results, with experimentation rates ranging from 15% to 23.5%. However, it is important to highlight the scarcity of data on this topic in the country, especially among school populations.

Regarding substances, cannabis remains the most widely used drug in Brazil and Latin America [16,17], followed by cocaine and amphetamines [7,12,16,17]. When stratified by sex, males present a higher lifetime prevalence of use [11,16,17,18].

Historically, drug abuse and dependence have been addressed reactively, often linked to social exclusion and criminality. However, effective approaches require the consideration of the social, psychological, economic, and political dimensions of this phenomenon [19]. Among adolescents, the distinct influences on initiation and continuation of illicit substance use are noteworthy. Bandura’s Social Learning Theory explains how adolescents’ school and family social networks may promote substance use behaviors through observational learning and reinforcement of significant role models [20]. Complementarily, involvement with drugs is associated with an imbalance between vulnerability factors (e.g., low parental supervision, history of violence, school failure, and peer influence) and protective factors (e.g., positive school bonds, family support, and high self-esteem) [21].

Furthermore, the cultural effects of late modernity or liquid modernity, as proposed by Bauman [22], deserve emphasis. According to the author, contemporary youth live in a context marked by uncertainties, and fragile social and constant pressures related to performance and consumption. This fluidity in relationships and identities, combined with the immediate pursuit of pleasure and relief from emotional tensions, may facilitate substance experimentation as a rapid coping or escape mechanism. Thus, adolescent drug use should not be understood merely as deviant behavior but as a response to the social, emotional, and cultural conditions in which young people are embedded.

Additionally, globalization and increased human mobility have intensified the expansion of illicit drug markets and, consequently, the ease of access. Although the COVID-19 pandemic temporarily reduced adolescent drug access due to social isolation and increased family supervision [23], the post-pandemic period has exacerbated social inequalities, such as unemployment and poverty, creating conditions conducive to both drug use and trafficking [1,3].

In this context, understanding the factors and motivations that lead adolescents to use psychoactive substances is crucial to support more effective prevention strategies. Therefore, the present study aims to determine the prevalence of drug experimentation and current use among high school students in the Metropolitan Region of Grande Vitória, Brazil, and to identify sociodemographic factors associated with these behaviors.

## 2. Materials and Methods

This cross-sectional analytical study was conducted in 2023 with students aged 14 to 19 from 63 public and private schools in the Grande Vitória Metropolitan Region (RMGV), Espírito Santo, which had an estimated population of about 1,880,828 in 2022 [24]. The RMGV encompasses seven municipalities: Cariacica, Fundão, Guarapari, Serra, Viana, Vila Velha, and Vitória.

The sample was selected using cluster sampling, with schools as the primary sampling units, stratified by municipality and school type (public/private; with or without participation in the “Estado Presente” Program), in order to ensure comparative representativeness across three strata: (1) public schools in areas participating in the program; (2) public schools in areas not participating in the program; and (3) private schools. The Estado Presente Program is a local state public policy aimed at violence prevention and combating school-related crime through the integration of social actions and the strengthening of citizenship in regions with high social vulnerability. The inclusion of schools both covered and not covered by the program aimed to support the monitoring and development of public policies aligned with its objectives.

A maximum margin of error of 5% was adopted for the overall sample and a margin of 10% per municipality. The total sample size was 4416 students, based on an initial proportion of 0.5 to maximize sample variance, according to Cochran’s methodology [25]. The response rate was exceptionally high.

Within each stratum, schools were selected using simple random sampling. After selecting the schools, classes were randomly drawn, as needed, to meet the calculated sample size. Students attending the evening shift were not included in the selection. The inclusion criteria were (1) being enrolled in high school in one of the selected schools; (2) being between 14 and 19 years old; and (3) submitting the Informed Consent Form (ICF) signed by parents or guardians, as well as the Assent Form (AF) signed by the adolescent. The exclusion criteria were (1) refusal to participate; (2) absence on the day of data collection; (3) questionnaires with more than 50% missing data, or duplicate questionnaires; (4) inconsistent or entirely blank questionnaires; or (5) students who did not return the signed forms.

The data collection instrument was based on questionnaires previously used in studies with university students from the University of São Paulo by Andrade et al. (1997) [26] and Stempliuk et al. (2005) [27], in addition to the National School Health Survey (PeNSE, 2019) [12]. Although Andrade and Stempliuk’s questionnaires were originally applied to older populations, the PeNSE instruments are validated for Brazilian adolescents. Furthermore, a pilot test was conducted in February 2023 in a non-selected school to test the feasibility and understanding of the instrument. Minor adjustments were made to the language to make it more accessible to high school students, with no need for translation, as the questionnaire was already in Portuguese.

After approval from the State Department of Education (SEDU), the selected schools were contacted via email and phone to schedule a presentation seminar, which took place in the auditorium of the Health Sciences Center (CCS) in March 2023. Approximately 80% of the schools participated in person. For those who did not attend, an individual visit was scheduled for a presentation of the study and delivery of the forms. In all schools, students received the ICFs, which were to be taken home and signed by their guardians.

Data collection took place between March and December 2023 by a trained and supervised field team. Prior to fieldwork, all team members received training using a specific manual and standardized instructions for the data collection procedure. Data were collected using tablets in an environment that ensured participants’ privacy. The questionnaire was self-administered, but researchers were present to clarify technical questions without influencing responses.

The dependent variable (outcome) included lifetime and current use of drugs in general, except alcohol and tobacco. It includes marijuana, crack, snorting products, substances to feel high, heroin, LSD, ecstasy, or methamphetamine. The question asked during the screening for experimentation was “Have you ever used/tried/snorted any of the drugs listed in your lifetime?” and, for the screening of current use, it was “Did you use/try/snort any of the drugs listed in the past 30 days?”

The independent variables (exposure) corresponded to the sociodemographic characteristics: sex (female or male); age in years (14 to 15 years, 16 to 17 years, and 18 to 19 years); sexual orientation (heterosexual and LGBTQIAPN+ population); gender identity (cisgender and non-cisgender); race/color (White, Black, and Brown); marital status (with or without a partner); socioeconomic class (A, B1/B2, C1/C2, D, and E); type of school (public or private); current employment status (yes or no); religion (catholic, evangelical, none, and other); and marital status of parents—living together (yes or no).

In Brazil, racial classification follows the categories established by the Brazilian Institute of Geography and Statistics (IBGE) [28], based on self-identification into five groups: White; Black; Brown (pardo); Asian; and Indigenous. Students who self-identified as Asian or Indigenous were excluded from the comparative analyses due to low sample representativeness. While methodologically justified, this exclusion represents a limitation regarding the ethnic–racial representativeness of the results and should be taken into account when interpreting and generalizing the findings. The Brazilian Economic Classification Criteria by ABEP was used to estimate financial value and the purchasing power of the Brazilian urban population, classifying people from A (highest income) to E (lowest income) [29].

Statistical analyses were performed with Stata 17.0: descriptive with absolute and relative frequencies and 95% confidence intervals (95%CI), bivariate with Pearson’s Chi-square (χ^2^), and multivariate using prevalence ratio (PR) via Poisson regression with robust variance. Variables with *p* ≤ 0.20 in the bivariate were entered into the model, keeping those with *p* ≤ 0.05. This study was approved by the UFES Ethics Committee (no. 5900370) following resolution 466/2012, and all participants signed Free and Informed Consent and Assent Forms.

## 3. Results

The following section presents the results of this study, including the prevalence of lifetime and current use of illicit substances—excluding alcohol and tobacco—as well as their associations with sociodemographic and behavioral variables.

Among the 4416 students surveyed, 1007 reported having used drugs at some point in their lives, corresponding to approximately 22% of the sample (95% CI: 20.7–23.1). Of these, 23.5% were girls and 19.8% were boys. Moreover, among all the adolescents screened, 8.7% (95% CI: 7.9–9.5) reported that they currently use some of the substances screened (N = 400), with this pattern being higher among girls (9.2% 95% CI: 8.1–10.4) when compared to boys (8.0% 95% CI: 6.9–9.2) (Table 1).

Among the motivations for experimenting with illicit drugs among high school students, patterns emerged that reflect both individual and social–emotional factors. As shown in Table 2, the most frequently reported reasons were to reduce stress (43.0%; 95% CI: 40.0–46.1), to relax (35.5%; 95% CI: 32.6–38.5), to have fun with friends (33.2%; 95% CI: 30.3–36.2), and to forget problems (29.1%; 95% CI: 26.4–32.0).

Emotional well-being–related issues were also frequently reported, such as “to feel good” (25.2%), “to relieve depression” (19.7%), and “to help with sleep” (15.2%). Other motivations, including social curiosity, group integration (5.9%), boredom (11.9%), and self-perception of dependence (2.3%), appeared in smaller proportions (Table 2).

It is also noteworthy that 32.3% of participants selected “none of the above,” which may reflect motivations not captured by the questionnaire or possible difficulties in understanding the question or identifying their own reasons. These findings suggest that substance use is often linked to emotional relief and the regulation of negative affective states (Table 2).

In the bivariate analysis, drug experimentation and current use were associated with nearly all socioeconomic factors examined, including age, gender identity, sexual orientation, race/color, marital status, school type, employment, religion, and parents’ marital status. Additionally, lifetime drug use was significantly associated with sex. Current drug use was also linked to socioeconomic class (*p* < 0.05). These data are presented in Table 3 below.

After adjusting for confounders, adolescents aged 18–19 had a 44% (95% CI: 1.18–1.74) higher prevalence of drug experimentation than younger peers, and LGBTQIAPN+ individuals had a 54% (95% CI: 1.36–1.75) higher frequency than heterosexuals. Black participants had 1.33 (95% CI: 1.14–1.54) times higher prevalence; those with a partner, 1.41 (95% CI: 1.26–1.58) times; and employed adolescents, 1.36 (95% CI: 1.21–1.53) times. Social class A showed a 1.24 (95% CI: 1.05–1.46) prevalence ratio compared to classes C/D/E. Adolescents whose parents no longer lived together had 1.29 (95% CI: 1.15–1.45) times higher prevalence than those with cohabiting parents (Table 4).

According to the adjusted analysis, being LGBTQIAPN+ increased the prevalence of current drug use by 1.76 (95% CI: 1.41–2.19) times, as did identifying as Black, which increased it by 1.59 times (95% CI: 1.22–2.06). Having a partner (PR: 1.42; 95% CI: 1.16–1.73), having a job or business (PR: 1.62; 95% CI: 1.32–1.98), and parents not living together (PR: 1.37; 95% CI: 1.12–1.68) presented higher frequencies of current drug use (Table 5).

The findings reveal significant patterns of illicit drug experimentation and current use among high school adolescents, particularly among older students, individuals from socially marginalized groups, and those with greater financial independence and lower family stability.

## 4. Discussion

Drug experimentation among high school students in the metropolitan region of Espírito Santo was approximately 22%, a slightly higher value than that found in the 2019 PENSE in the municipality of Vitória. In that survey, it was recorded that 16.6% (95% CI: 12.5–20.8) of elementary school students had used drugs at some point in their lives [11]. In other Brazilian studies, the prevalence found varied between 22% and 24% [2,13,16], which is close to the data found in this study. In a global analysis of adolescents from 47 countries, prevalence rates above 20% were observed in countries such as the USA and France, while Southeast Asian countries, such as Malaysia, have significantly lower rates [17,30].

Regarding current drug use, the frequency found was 8.7%, which is almost the same as that found in Vitória in 2019 (9.5%), remaining higher than the national average, which was recorded at 5.1% [12]. The World Drug Report mentions that the use of new psychoactive substances is generally higher among school students than among the general population and estimates that in South America, more than half of the people undergoing drug treatment are under 25 years of age [3].

The adolescence phase has numerous complexities related to hormonal, psychological, relational, and social issues that interfere with various developmental processes, especially issues related to health and adherence to risky behaviors, such as experimentation with alcohol, cigarettes, and other drugs [9]. Studies show that, in general, drug experimentation begins at this stage and may or may not extend into adulthood [3].

The motivation behind adolescent drug use is a key aspect in preventing this behavior. This research identified the influence of emotions and personal relationships, with students reporting drug use to reduce stress, relax, and have fun with friends. As already discussed in the Introduction, adolescent substance use should not be interpreted solely as deviant or irrational behavior, but as a response to complex emotional, social, economic, and cultural circumstances.

Family and school social networks may contribute to the normalization of substance use through observational learning, as proposed by Bandura’s Social Learning Theory [20,21]. Additionally, the contemporary scenario described as “liquid modernity”—marked by instability, fragile bonds, and constant performance demands—fosters drug experimentation as a quick coping strategy [22].

The pursuit of pleasure may indicate symptoms of depression and anxiety, with substances used as a coping strategy for internal and external conflicts [30,31]. This pattern can be interpreted as a dysfunctional emotional regulation strategy, consistent with the self-medication theory, which suggests that adolescents experiencing psychological distress may turn to drugs to alleviate their discomfort [30,31]. The anticipated psychoactive effects of illicit substances, such as altered states of consciousness, along with the natural curiosity characteristic of this stage of development, also play a relevant role [4,9].

Other studies also report pleasure-seeking as a motivation for drug use among students and their peers [32,33,34,35], and in São Paulo, an association was identified between drug use and having friends who used, sold, or brought drugs to parties [33].

From a neurobiological perspective, the immaturity of the prefrontal cortex—which regulates inhibitory control and decision-making—combined with the hyperactivity of the reward system, makes adolescents more prone to impulsive and risky behaviors, such as drug use. Peer influence during this developmental stage is also reinforced by neural mechanisms related to social acceptance and dopaminergic reinforcement, intensifying the likelihood of experimentation [4,9].

Currently, peer groups—especially those formed in school settings—play a central role in adolescent socialization, sometimes surpassing the influence of family. These groups are instrumental in shaping identity and subjectivity and can exert both protective and risk-enhancing influences, as is the case with drug use [31,36,37]. This underscores the importance of examining substance use in school contexts, which represent strategic environments for adolescent socialization and experimentation.

Another point of interest was the assessment of sex in relation to drug use. Drug experimentation was identified for females at around 24% and for boys at around 20%, as well as a higher frequency among girls for those who reported current drug use. Data from Brazil indicate slightly lower values, where experimentation among girls in 2019 represented 13.1% (95% CI: 10.9–15.3) of prevalence, while boys presented 11.0% (95% CI: 9.6–12.4) [12]. However, when evaluating the municipality of Vitória in 2010, similar values were found and with greater frequency among girls (25.1%) in relation to boys (21.7%) [16].

It is essential to recognize that patterns of drug use vary not only according to sex assigned at birth but also by gender identity, which encompasses individual self-perception and expression. Although this study considered sex as a binary variable (male/female), we acknowledge the limitations of this approach and emphasize the importance of future research adopting more inclusive designs that account for gender diversity to better capture these nuances.

Furthermore, it is necessary to evaluate different types of drugs and usage patterns between sexes, considering individuality, personality, and genetic factors. Studies indicate higher alcohol consumption among girls and higher tobacco use among boys in the case of legal drugs [30]. Regarding illicit drugs, marijuana, cocaine, and crack are more commonly used by males [2,12], while females show a higher prevalence of anxiolytic and amphetamine use [2]. These patterns may be influenced by both biological and social determinants. Biologically, hormonal fluctuations in females may increase sensitivity to the dopaminergic effects of certain drugs, potentially enhancing their susceptibility [9,38]. Socially, gender norms and expectations may reinforce unequal exposure to risk factors, such as interpersonal violence and trauma, which disproportionately affect girls and women and may contribute to substance use as a coping mechanism [3]. Given these complexities, future research should adopt a broader, more inclusive framework that distinguishes between sex and gender identity, ensuring scientifically rigorous analyses that reflect the diverse experiences of adolescents.

In addition, it was found that older adolescents, between 18 and 19 years old, exhibited a higher prevalence of drug experimentation compared to younger adolescents. In line with this finding, the Brazilian Drug Report also points to greater experimentation among older adolescents and highlights that alcohol consumption is common even among younger ones [2]. A Mexican study found a pattern of drug experimentation among young people related to increasing age according to the type of substance used. The average age of initiation was 16.7 years for alcohol, 16.8 years for tobacco, 17 years for marijuana, and 17.3 years for cocaine [39]. The literature describes a pattern of substance use that varies over time. Notably, most adolescents begin with alcohol and cigarettes, and as the years pass, new substances are introduced, such as marijuana, cocaine, and even crack, which may justify the stronger association of illicit drug use with older adolescents [2,39,40].

Adolescents identifying as LGBTQIAPN+ had a 54% higher lifetime drug use than heterosexuals. Vulnerable groups like LGBTQIAPN+ individuals may be more susceptible due to specific factors. In the U.S., a national survey showed illicit drug misuse was 2 to 3 times higher among LGBTQIAPN+ adults compared to heterosexuals [41]. Another U.S. study (2005–2017) found significantly elevated drug use among sexual minorities of both sexes for all drug types [9]. The Minority Stress Theory explains this vulnerability by highlighting stress from discrimination and intolerance [42].

A significant association was found between Black skin color and both experimentation and current drug use. In Brazil, racial identification is deeply intertwined with socioeconomic inequalities resulting from the enduring legacy of slavery, racial segregation, and structural racism, which shape unequal access to rights, opportunities, and health outcomes.

National surveys show that most crack users self-identify as Black or Brown [2], while the use of substances such as methamphetamine and heroin remains more prevalent among White populations in other countries [43]. However, such patterns cannot be understood solely by racial categorization, but rather through the intersection of race, class, territory, and access to public policies. Black and Brown populations in Brazil are disproportionately exposed to poverty, under-resourced schools, urban violence, and weak health and social protection networks—conditions that increase vulnerability to substance use and reduce access to care and rehabilitation services [43,44,45].

Moreover, racial disparities are evident in the differential treatment of drug users by the criminal justice and health systems. Studies show that Black individuals are more likely to be arrested and less likely to receive support through therapeutic or harm reduction programs when compared to White individuals under similar circumstances [44,45]. These findings reveal how the consequences of drug use are not only biological or behavioral but also socially produced and unequally distributed, demanding the adoption of culturally sensitive, anti-racist approaches in both prevention and treatment.

The marital status of students also showed statistical significance. Those who reported having a partner were more likely to have used drugs at some point in their lives and currently. Although this study assessed sex as a binary variable (male/female) and did not stratify results based on gender identity, it is important to recognize that the influence of romantic relationships on drug use may be mediated by both biological sex and socially constructed gender roles. In contrast to this analysis, a study conducted with young people about substance dependence identified dating as a positive influence on the life of the adolescent, acting as a coping mechanism and helping them to distance themselves from family struggles, making their relationship a form of support and trust [31]. Another study carried out with women portrays relationships as a possible measure of risk for drug use and abuse. This is due to the idealization of the relationship and the expectation of life transformation through a partner. If frustrations about this role arise, there is a need to seek pleasure through drugs. Another issue is the influence of partners who were already drug users because, in order to maintain the relationship, women started using substances [46]. As can be seen, the gender issue greatly influences this variable, bringing a need for a more extensive assessment, especially regarding adolescents.

These findings illustrate how gender-related dynamics—whether derived from sex-based biological vulnerabilities or from gender identity, social roles, and relationship patterns—play a crucial role in shaping adolescent behavior. Therefore, a more comprehensive understanding of how gender identity intersects with romantic experiences is essential. Future research should aim to differentiate between sex and gender more explicitly and consider inclusive methodologies that reflect the diversity of adolescent experiences.

Adolescents who reported having a job showed higher drug use prevalence: PR of 1.36 (95% CI: 1.21–1.53) for experimentation and 1.62 (95% CI: 1.32–1.98) for current use. Although adolescent work can, in some cases, promote responsibility and autonomy, in the Brazilian context, it is often associated with socioeconomic vulnerability and precarious labor conditions. Many adolescents enter the workforce not as a developmental choice but out of necessity, frequently to contribute to household income due to financial hardship. This dynamic leads to the early assumption of adult roles and responsibilities, such as long working hours, economic stress, and emotional burden, which can interfere with schooling, psychosocial development, and mental health [47].

This premature adultification may lead to feelings of exhaustion, frustration, and anxiety, contributing to the adoption of maladaptive coping mechanisms, including substance use. Evidence suggests that adolescents who balance school and work are at greater risk of school dropout, absenteeism, and disengagement—factors that are, in themselves, associated with increased vulnerability to drug use [13]. These young individuals often have limited access to leisure activities, cultural engagement, and psychosocial support—key protective factors during adolescence, which heightens their exposure to stressors [9,48]. In Brazil, the Child and Adolescent Statute (ECRIARDE) offers important guidelines to protect these adolescents [48].

Furthermore, individuals from social class A presented a prevalence ratio of 1.24 times higher in relation to individuals from classes C/D/E, similar to that found in a survey in the Midwest, in which individuals from high and medium economic levels presented 1.45 times more frequency of drug use (95%CI 1.10; 1.92) [13]. In Vitória, this pattern was also observed when analyzing the higher lifetime drug use among students enrolled in private schools, where the majority belonged to social classes A and B. The increase in the source of income is associated with greater purchasing power for certain substances, creating broader accessibility to the illicit drug market [16].

It is crucial to recognize that adolescents from different socioeconomic backgrounds not only engage in substance use for different reasons but also experience unequal consequences, reflecting broader social inequities. Thus, interpretations must go beyond material conditions to consider the symbolic and institutional factors that shape these patterns.

Adolescents reporting that their parents no longer lived together showed a significant association with lifetime and current drug use. In Mato Grosso, drug use was linked to unsatisfactory parental relationships (PR: 1.43; 95% CI: 1.08–1.91) [13]. Another study found higher substance use among those whose parents were separated, divorced, or deceased compared to those living with both parents [31]. Family conflicts and mourning in these situations affect adolescents’ emotional health, increasing vulnerability to drug use [31].

This study employed a cross-sectional design, which precludes causal inferences. Additionally, there is a potential risk of information bias due to the self-reported nature of the data. Nevertheless, the confidential and individual administration of the questionnaire helped to minimize potential discomfort and enhanced the reliability of the responses. Other limitations include the lack of control for psychiatric comorbidities such as depression and anxiety; insufficient exploration of gender identity categories; and the absence of detailed stratification of drug types based on psychosocial variables. Future longitudinal studies with representative samples are warranted to address these gaps.

## 5. Conclusions

Drug experimentation and current use among high school adolescents in Espírito Santo’s Metropolitan Region were highly prevalent. Adolescents aged 18–19; those identifying as LGBTQIAPN+, Black, or from higher economic classes; as well as those with a job, a partner, or whose parents no longer lived together, showed higher rates of drug use. These findings highlight the influence of social, cultural, and relational factors on drug use.

Public health policies must therefore move beyond universal, punitive, or reactive approaches, prioritizing intersectional and equity-based strategies. Preventive actions should be culturally sensitive and tailored to specific vulnerabilities—such as those faced by LGBTQIAPN+ youth, Black adolescents, and students in both high- and low-income settings—fostering resilience through school engagement, family support, and access to psychosocial care. Programs should also address the impacts of adultification, precarious labor, and relational stressors in adolescent life.

From a public health perspective, it is crucial to expand school-based mental health services, implement inclusive and evidence-informed health education, offer structured recreational opportunities for adolescents, and ensure targeted support for youth engaged in paid work. Future research agendas should further investigate the intersections between substance use, identity, emotional regulation, and structural determinants, employing longitudinal designs and participatory methodologies. Moreover, intersectoral collaboration between health, education, and social protection systems is essential to foster supportive environments that mitigate risk and promote adolescent well-being.

## Figures and Tables

**Table 1 ijerph-22-01373-t001:** Lifetime and current drug use among high school students in the Greater Vitória Metropolitan Region, Espírito Santo, from March to December 2023.

Variables	General Sample			Female			Male		
	N	%	95%CI	N	%	95%CI	N	%	95%CI
use of any drug in life, except alcohol and tobacco *									
No	3609	78.2	77.0–79.3	1917	76.5	74.8–78.1	1689	80.2	78.4–81.8
Yes	1007	21.8	20.7–23.1	588	23.5	21.9–25.2	418	19.8	18.2–21.6
current use of any drug, except alcohol and tobacco *									
No	4216	91.3	90.5–92.1	2274	90.8	89.6–91.9	1939	92.0	90.8–93.1
Yes	400	8.7	7.9–9.5	231	9.2	8.1–10.4	168	8.0	6.9–9.2

N: absolute frequency; %: relative frequency; 95%CI: 95% confidence interval; * 1 missing data entry (in sex).

**Table 2 ijerph-22-01373-t002:** Reasons for experimenting with illicit drugs among high school students in the Grande Vitória Metropolitan Region, Espírito Santo, from March to December 2023.

	N	%	95%CI
Among those who tried crack, snorted products to feel high, marijuana, cocaine, heroin, LSD, ecstasy, or methamphetamine throughout life, what motivation(s) do you consider to be the most important for your use of drugs? (N = 1007)			
to reduce stress	433	43.0	40.0–46.1
to have fun with friends	334	33.2	30.3–36.2
to fit in with the group I belong to	59	5.9	4.6–7.5
to forget my problems	293	29.1	26.4–32.0
to not feel bored	120	11.9	10.1–14.1
to feel good	254	25.2	22.6–28.0
to relieve depression	198	19.7	17.3–22.2
to be able to sleep	153	15.2	13.1–17.5
to increase the chances of sexual encounters	44	4.4	3.3–5.8
to celebrate important occasions	84	8.3	6.8–10.2
because I get funnier	125	12.4	10.5–14.6
to relax	357	35.5	32.6–38.5
because it is easier to talk to people	76	7.6	6.1–9.4
because I believe I am dependent	23	2.3	1.5–3.4
because everyone uses it	53	5.3	4.0–6.8
none of the above	325	32.3	29.5–35.2

N: absolute frequency; %: relative frequency; 95%CI: 95% confidence interval.

**Table 3 ijerph-22-01373-t003:** Distribution of drug use among high school students in the Grande Vitória Metropolitan Region, Espírito Santo, from March to December 2023. N = 1007.

Variables	General Drug Experimentation				Current General Drug Use			
	N	%	95%CI	*p*-Value	N	%	95%CI	*p*-Value
Sex * (N = 1006)				0.003				0.133
female	588	23.5	21.9–25.2		231	9.2	8.1–10.4	
male	418	19.8	18.2–21.6		168	8.0	6.9–9.0	
Age (years) * (N = 1006)				<0.001				0.012
14 to 15	168	17.3	15.1–19.8		66	6.8	5.4–8.6	
16 to 17	668	22.0	20.6–23.5		266	8.8	7.8–9.8	
18 to 19	170	28.2	24.7–31.9		67	11.1	8.8–13.9	
Gender identity ** (N = 1005)				0.001				<0.001
cisgender	919	21.3	20.1–22.5		345	8.2	7.4–9.1	
non-cisgender	86	29.5	24.5–34.9		45	15.4	11.7–20.0	
Sexual orientation * (N = 1006)				<0.001				<0.001
heterosexual	687	18.8	17.5–20.1		255	7.0	6.2–7.8	
LGBTQIAPN+	319	33.6	30.7–36.7		144	15.2	13.0–17.6	
Race/skin color *** (N = 963)				<0.001				<0.001
White	355	20.0	18.2–21.9		128	7.2	6.1–8.5	
Black	214	28.2	25.1–31.5		95	12.5	10.3–15.0	
Brown	394	20.2	18.5–22.1		155	8.0	6.8–29.4	
Marital status				<0.001				<0.001
with partner	364	29.6	27.2–32.3		251	7.4	6.6–8.3	
no partner	643	19.0	17.7–20.4		149	12.1	10.4–14.1	
Socioeconomic classification ** (N = 1005)				0.053				0.044
A	185	24.3	21.3–27.4		74	9.7	7.8–12.0	
B1/B2	443	20.3	18.7–22.1		180	8.3	7.2–9.5	
C1/C2/D/E	377	22.6	20.7–24.7		146	8.8	7.5–10.2	
Type of school				<0.001				<0.001
public	721	23.3	21.9–24.9		301	9.7	8.7–10.8	
private	286	18.8	16.9–21.8		99	6.5	5.4–7.9	
Do you currently have any work, employment, or business?				<0.001				<0.001
no	683	19.5	18.3–20.9		261	7.5	6.6–8.4	
yes	324	29.0	26.4–31.7		139	12.4	10.6–14.5	
Religion				<0.001				<0.001
none	322	29.6	26.9–32.3		137	12.6	10.7–14.7	
catholic	234	19.9	17.7–22.2		90	7.6	6.3–9.3	
evangelical	343	17.1	15.5–18.8		118	5.9	4.9–7.0	
others	108	31.9	27.2–37.1		55	16.3	12.7–20.6	
Marital status of parents—do they live together? *** (N = 999)				<0.001				<0.001
yes	436	18.0	16.5–19.6		159	6.6	5.7–7.6	
no	563	26.0	24.2–27.9		235	10.9	9.6–12.2	

N: absolute frequency; %: relative frequency; 95%CI: 95% confidence interval; * 1 missing data entry; ** 2 missing data entries; *** 8 missing data entries.

**Table 4 ijerph-22-01373-t004:** Crude and adjusted odds ratios of drug experimentation among high school students in the Grande Vitória Metropolitan Region, Espírito Santo, from March to December 2023.

Variables	General Drug Experimentation					
	Gross RP	95%CI	*p*-Value	Adjusted PR	95%CI	*p*-Value
Sex			0.003			0.076
female	1.18	1.06–1.32		1.11	0.99–1.25	
male	1.0			1.0		
Age (years)			<0.001			0.001
14 to 15	1.0			1.0		
16 to 17	1.27	1.09–1.48		1.23	1.05–1.44	
18 to 19	1.63	1.35–1.96		1.44	1.18–1.74	
Gender identity			0.001			0.988
cisgender	1.0			1.0		
non-cisgender	1.38	1.15–1.67		1.02	0.82–1.22	
Sexual orientation			<0.001			<0.001
heterosexual	1.0			1.0		
LGBTQIAPN+	1.79	1.60–2.00		1.54	1.36–1.75	
Race/skin color			<0.001			<0.001
White	1.0			1.0		
Black	1.41	1.22–1.63		1.33	1.14–1.54	
Brown	1.01	0.89–1.15		1.04	0.91–1.18	
Marital status			<0.001			<0.001
with partner	1.56	1.40–1.74		1.41	1.26–1.58	
no partner	1.0			1.0		
Socioeconomic classification			0.052			0.015
A	1.07	0.92–1.25		1.24	1.05–1.46	
B	0.90	0.80–1.02		0.99	0.88–1.12	
C/D/E	1.0			1.0		
Type of school			0.001			0.084
public	1.24	1.10–1.40		1.13	0.98–1.31	
private	1.0			1.0		
Do you currently have any work, employment, or business?			<0.001			<0.001
no	1.0			1.0		
yes	1.48	1.32–1.66		1.36	1.21–1.53	
Religion			<0.001			<0.001
none	1.0			1.0		
catholic	0.67	0.58–0.78		0.81	0.82–1.20	
evangelical	0.58	0.51–0.66		0.66	0.57–0.77	
others	1.08	0.90–1.29		0.99	0.69–0.95	
Marital status of parents—do they live together?			<0.001			<0.001
yes	1.0			1.0		
no	1.44	1.27–1.64		1.29	1.15–1.45	

RP: Prevalence Ratio; 95%CI: 95% confidence interval.

**Table 5 ijerph-22-01373-t005:** Crude and adjusted analysis of current drug use among high school students in the Grande Vitória Metropolitan Region, Espírito Santo, from March to December 2023.

Variables	Current General Drug Use					
	Gross RP	95%CI	*p*-Value	Adjusted PR	95%CI	*p*-Value
Sex			0.134			0.361
female	1.16	0.96–1.40		1.10	0.90–1.35	
male	1.0			1.0		
Age (years)			0.013			0.272
14 to 15	1.0			1.0		
16 to 17	1.29	0.99–1.67		1.20	0.91–1.57	
18 to 19	1.63	1.18–2.26		1.31	0.93–1.85	
Gender identity			<0.001			0.296
cisgender	1.0			1.0		
non-cisgender	1.88	1.41–2.50		1.19	0.86–1.67	
Sexual orientation			<0.001			<0.001
heterosexual	1.0	1.80–2.64		1.0		
LGBTQIAPN+	2.18			1.76	1.41–2.19	
Race/skin color			<0.001			0.001
White	1.0			1.0		
Black	1.73	1.35–2.23		1.59	1.22–2.06	
Brown	1.10	0.88–1.38		1.11	0.88–1.39	
Marital status			<0.001			0.001
no partner	1.0			1.0		
with partner	1.64	1.35–1.98		1.42	1.16–1.73	
Socioeconomic classification			0.474	-	-	-
A	0.94	0.77–1.16		-	-	
B1/B2	1.11	0.85–1.45		-	-	
C1/C2/D/E	1.0					
Type of school						0.158
public	1.50	1.20–1.86	<0.001	1.19	0.93–1.53	
private	1.0			1.0		
Do you currently have any work, employment, or business?			<0.001		1.0	<0.001
no	1.0			1.0		
yes	1.66	1.37–2.02		1.62	1.32–1.98	
Religion			<0.001			<0.001
none	1.0			1.0		
catholic	0.61	0.47–0.78		0.78	0.59–1.03	
evangelical	0.47	0.37–0.59		0.56	0.43–0.72	
others	1.29	0.97–1.73		1.12	0.32–1.53	
Marital status of parents—do they live together?			<0.001			0.002
yes	1.0			1.0		
no	1.65	1.35–2.02		1.37	1.12–1.68	

RP: Prevalence Ratio; 95%CI: 95% confidence interval.

## Data Availability

The original contributions presented in this study are included in the article. Further inquiries can be directed to the corresponding author.

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
