# Peer review of "Illicit Drug Use and Sociodemographic Correlates Among Adolescents in a Brazilian Metropolitan Region: A School-Based Cross-Sectional Study"

_ijerph, 2025, doi:10.3390/ijerph22091373_

Round 1

Reviewer 1 Report

Comments and Suggestions for Authors

See the attachment.

Comments on the Quality of English Language

As a non-native English speaker, I found the manuscript generally clear and comprehensible.

Author Response

Dear Reviewer,

We would like to sincerely thank you for the time and effort dedicated to reviewing our manuscript. We highly appreciate your valuable suggestions and constructive comments, which have significantly contributed to improving the quality and clarity of our work.
We are pleased to inform you that all your recommendations have been fully addressed. The corresponding modifications have been incorporated into the revised version of the manuscript, which is attached to this message. For your convenience, all changes have been highlighted in red within the text.
We are confident that these revisions have strengthened the manuscript and addressed all the points raised. Once again, we are grateful for your careful review and thoughtful feedback.

Reviewer 2 Report

Comments and Suggestions for Authors

Recommendation:

Accept with Major Revisions

Summary of the Article

This cross-sectional study investigates the prevalence and associated factors of illicit drug use among high school students in the Greater Vitória Metropolitan Region, Brazil. Using a structured questionnaire and statistical analysis (including Poisson regression), the authors identify sociodemographic variables linked to both lifetime and current drug use.

2. Strengths

  • Relevance and Timeliness: The topic is highly relevant to public health and adolescent well-being, particularly in Brazil.
  • Sample Size and Scope: The study includes a large, diverse sample from both public and private schools across multiple municipalities.
  • Ethical Considerations: Ethical approval and informed consent procedures are clearly described.
  • Statistical Rigor: The use of bivariate and multivariate analyses is appropriate and well-documented.

3. Areas for Development

a. Introduction

  • The introduction is too brief and lacks sufficient academic grounding, especially in the Brazilian context.
  • It does not adequately reference existing (national) studies on adolescent drug use in Brazil, 
  • Theoretical frameworks explaining why adolescents engage in drug use (e.g., Normalisation, role of night time economy, Social Learning Theory, Minority Stress Theory, or Risk and Protective Factors models, how modernity impacts adolescent and young adult pathways) are absent. Including these would help frame the problem more robustly.

b. Findings Section

  • The results section is overly reliant on data tables with minimal narrative explanation.
  • There is a lack of interpretive commentary to guide the reader through the significance of the findings.
  • The authors should highlight key trends, explain unexpected results, and connect findings to the literature.

c. Methodological Limitations

  • The exclusion of alcohol and tobacco from the analysis should be justified more clearly, as these are often gateway substances.
  • The limitations of cross-sectional design are acknowledged but not sufficiently discussed.

e. Terminology and Sensitivity

  • Terms like “non-cisgender” and “LGBT+” should be defined clearly and used consistently.
  • Discussions of race and socioeconomic status should be more nuanced and culturally sensitive.

f. Formatting and Referencing

  • Several references are outdated or inconsistently formatted.
  • The citation style should be standardized according to journal guidelines.

4. Specific Suggestions for Revision

  1. Expand the Introduction:

    • Include more academic Brazilian studies and statistics on adolescent drug use.
    • Introduce relevant theoretical frameworks to explain youth drug behaviour.
  2. Revise the Results Section:

    • Add narrative explanations to accompany tables.
    • Highlight key findings and their implications.
  3. Improve Language and Structure:

    • Conduct a professional language edit.
  4. Enhance the Discussion:

    • be specific about recommendations to policy, practice and research based on your findings
    • Compare findings with national/international literature.
    • Discuss limitations and implications more critically.
  5. Clarify Terminology:

    • Define demographic categories and use inclusive language.
  6. Update and Standardize References:

    • Ensure all citations are current and properly formatted.

5. Final Judgment

The study addresses an important issue and presents valuable data. However, major revisions are necessary to improve the manuscript’s academic rigor, clarity, and contextual relevance.

Comments on the Quality of English Language

.Writing and Language Quality

  • The manuscript contains numerous grammatical errors and awkward phrasing.
  • The abstract and introduction should be rewritten for clarity and conciseness.

Author Response

(The authors gave the same response as above.)

Reviewer 3 Report

Comments and Suggestions for Authors

The paper is generally well written, with a clear structure. However, there are some few points that need to be addressed:

  • Materials and Methods: This section needs more detail. In particular, the sample size should be explicitly stated. Additionally, the total number of participants is not clearly mentioned in the results section, which should be corrected for clarity and completeness.
  • Table 3: Please remove the use of capital letters in the table entries, as it is inconsistent with formatting standards.

Author Response

(The authors gave the same response as above.)
